# Why Can't You Do That HAL?
# Explaining Unsolvability of Planning Tasks

**Sarath Sreedharan[1], Siddharth Srivastava[1], David Smith[2] , Subbarao Kambhampati[1]**

[1]CIDSE, Arizona State University, Tempe, AZ 85281 USA

[2]PSresearch

ssreedh3@asu.edu, siddharths@asu.edu, david.smith@psresearch.xyz, rao@asu.edu

## Abstract

Explainable planning is widely accepted as a pre-requisite for autonomous agents to successfully work with humans. While there has been a lot of research on generating explanations of solutions to planning problems, explaining the absence of solutions remains a largely open and under-studied problem, even though such situations can be the hardest to understand or debug. In this paper, we show that hierarchical abstractions can be used to efficiently generate reasons for unsolvability of planning problems. In contrast to related work on computing certificates of unsolvability, we show that our methods can generate compact, human-understandable reasons for unsolvability. Empirical analysis and user studies show the validity of our methods as well as their computational efficacy on a number of benchmark planning domains.

## 1 Introduction

The ability to explain the rationale behind a decision is widely seen as one of the basic skills needed by an autonomous agent to truly collaborate with humans. At the very least we would want our autonomous assistants to be capable of explaining why a particular action/plan was chosen to achieve some objective and be able to explain why they consider some objectives to be unachievable. For example, consider an automated taxi scheduling system. A user asks for a taxi to pick up her and three of her friends and the service comes back by saying that it is not possible, and recommends instead using two different taxis. In this scenario, the user would want to know why a single taxi can't pick up all four of them.

Most earlier works in explanation generation for planning problems have focused on the problem of explaining why a given plan or action was chosen, but do not address the problem of explaining the unsolvability of a given planning problem. The few works that have tried to address unsolvability have mostly looked at generating certificates or proofs of unsolvability (cf. [Eriksson, Röger, and Helmert, 2018; 2017]) or identify some modification of the planning problem that could make the problem solvable, i.e, an excuse for the unsolvability of the problem (c.f [Göbelbecker et al., 2010]). Unfortunately the certificates/proofs considered by these works are geared towards automatic verification rather than human understandability and for complex domains excuses generated by such systems may not be enough to understand why a problem was unsolvable in the first place.

In this paper, we present a new approach for explaining unsolvability of planning problems that builds on the well known psychological insight that humans tend to decompose sequential planning problems in terms of the subgoals they need to achieve [Donnarumma, Maisto, and Pezzulo, 2016; Cooper and Shallice, 2006; Simon and Newell, 1971]. We will thus help the user understand the infeasibility of a given planning problem by pointing out unreachable but necessary subgoals. For example, in the earlier case, "Holding three passengers" is a subgoal that is required to reach the goal, but one that can no longer be achieved due to new city regulations. Thus the system could explain that the taxi can't hold more than two passengers at a time (and also notify the user about the new city ordinance).

Unfortunately, this is not so straightforward, since by the very nature of the problem, there exist no solutions and hence its hard to extract meaningful non-trivial subgoals for the problem. We can find a way around this issue by noting the fact that the user is asking for an explanation for unsolvability either due to a lack of understanding of the task or because of limitations in their inferential capabilities. Therefore, we can try to capture the user's expectations by considering abstractions of the given problem. In particular, we use state abstractions to generate potential solutions and subgoals at higher levels of abstractions. Such an approach was used by [Sreedharan, Srivastava, and Kambhampati, 2018] to compute explanations for user queries attuned to the level of expertise of the user.

In section 3, we present our basic framework and discuss how we can identify the appropriate level of abstraction and unachievable subgoals for an unsolvable classical planning problem. In the real world, a more challenging version of this problem arises when the user provides *plan advice* (which may include temporal preferences) on the type of solutions expected. In section 4, we will see how explaining unsolvability of planning problems with plan advice (c.f [Myers, 1996]) could be seen as establishing unsolvability of planning problems with additional plan constraints. This is a capability that is necessary to capture the fact that these explanations are being provided within the context of a conversation.

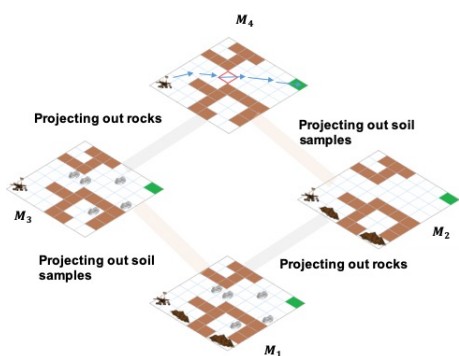

Figure 1: A sample abstraction lattice. The lattice consists of models generated by projecting out rocks or soil samples. Dark blobs are locations for soil samples, gray objects are rocks, and the goal is marked in green. The problem is unsolvable in the most concrete model but solvable in models where rocks are projected out.

The presence of this additional plan advice could either reflect cases (1) where the original problem was solvable, but the user's requirements (i.e. expressed in the advice) renders it unsolvable and (2) where the original problem was unsolvable and the user presents an outline for a solution in the form of advice. Even in the second case, by taking into account the human's expected solution, we can provide a more targeted explanation. Such additional advice are quite common within the context of contrastive explanations [Miller, 2018], where these advice specify alternative *foils* (in this case alternative plans) expected by the user. By supporting refutation of such advice we also allow the possibility of leveraging our approach for contrastive explanations. For evaluating our approach, we will present a user study we ran to validate the usefulness of such explanations for unsolvable problems (with plan advice) and also note the computational efficiency of our method for some standard planning benchmarks.

## 2 Background

We will assume that the autonomous agent uses a STRIPS planning model [Fikes and Nilsson, 1971] that can be represented as a tuple of the form $\mathcal{M} = \langle F, A, I, G \rangle$, where $F$ is a set of propositional fluents that define the state space $\mathbb{S}_M$ for the model, $A$ gives the set of actions the robot has access to, $I$ defines the initial state and $G$ the goal. A state $S \in \mathbb{S}_{\mathcal{M}}$ corresponds to a unique value assignment for each state fluent and can be represented by the set of fluents that are true in that state. Each action $a \in A$ is further defined by a tuple $a = \langle \text{prec}^a, \text{adds}^a, \text{dels}^a \rangle$ and a plan is defined as an action sequence of the form $\pi = \langle a_1, ..., a_n \rangle$. A plan is said to be valid for $\mathcal{M}$, if the result of executing a plan from the initial state satisfies the goal (denoted as $\pi(I) \models_{\mathcal{M}} G$). For the model $\mathcal{M}$, we will represent the set of all valid plans as $\Pi_{\mathcal{M}}$. Each planning model $\mathcal{M}$ also corresponds to a transition system $\mathcal{T} = \langle \mathbb{S}_{\mathcal{M}}, I, \mathbb{S}_G, A, T \rangle$, where $\mathbb{S}_G$ is the subset of $\mathbb{S}_{\mathcal{M}}$ where the goal $G$ is satisfied and $T \subseteq \mathbb{S}_{\mathcal{M}} \times A \times \mathbb{S}_{\mathcal{M}}$, such that $\langle S, a, S' \rangle \in T$ (denoted as $S \xrightarrow{a} S'$) if $S \subseteq \text{prec}^a$ and $a(S) = S'$. Each valid plan has a corresponding path in the

transition system from I to some state in $\mathbb{S}_G$.

In this work, we will be focusing on state and action abstractions induced by projecting out fluents. Thus a model $\mathcal{M}_2$ is said to be an abstraction of $\mathcal{M}_1$ (denoted by $\mathcal{M}_1 \sqsubseteq \mathcal{M}_2$) if model $\mathcal{M}_2$ can be formed from $\mathcal{M}_1$ by projecting out a set of fluents. Formally, $\mathcal{M}_1 \sqsubseteq \mathcal{M}_2$ if there exists some $P \subseteq F$, such that the transition system of $\mathcal{M}_2$ is defined as $\mathcal{T}_2 = \langle \mathbb{S}_{\mathcal{M}_2}, I_2, \mathbb{S}_{G_2}, A, T_2 \rangle$. Where, for every $S \in \mathbb{S}_{\mathcal{M}_1}$, there exist a state $S \setminus P \in \mathbb{S}_{\mathcal{M}_2}$, $I_2 = I \setminus P$, $\mathbb{S}_{G_2}$ is the subset of $\mathbb{S}_{\mathcal{M}_2}$ that satisfy $G' = G \setminus P$ and for every transition $\langle S, a, S' \rangle \in T_1$, there exist $\langle S \setminus P, a, S' \setminus P \rangle \in T_2$. We will denote an abstraction formed by projecting out $P$ from the model $\mathcal{M}$ as $f_P(\mathcal{M})$. An abstraction $f_P(\mathcal{M})$ is considered logically complete if for every $\pi$ such that $\pi(I) \models_{\mathcal{M}} G$, we have $\pi(I_{f_P(\mathcal{M})}) \models_{f_P(\mathcal{M})} G_{f_P(\mathcal{M})}$. In this work, we will only be looking at logically complete abstractions. For classical planning models, logically complete abstractions can be formed by simply removing the abstracted out fluents from the domain model and problem descriptions.

Sreedharan, Srivastava, and Kambhampati (2018) note that given a model $\mathcal{M}$ and a set of propositions $P$ we can define an abstraction lattice, denoted as $\mathbb{L}_{\mathcal{M},P} = \langle \mathbb{M}, \mathbb{E}, \ell \rangle$, where each model in $\mathbb{M}$ is an abstraction of $\mathcal{M}$ formed by projecting out some subset of fluents from $P$ (where $P \subseteq F$) and $\mathcal{M} \in \mathbb{M}$. There exist an edge $\langle \mathcal{M}_1, \mathcal{M}_2 \rangle \in \mathbb{E}$ with label $\ell(\mathcal{M}_1, \mathcal{M}_2) = p$, if $f_{\{p\}}(\mathcal{M}_1) = \mathcal{M}_2$, thus this structure provides a way of capturing the ordering induced by $\sqsubseteq$ (where elements higher up in the lattice are more abstract than the ones at the bottom). Note that in the most general case, the lattice need not be complete, that is $|\mathbb{M}| \neq 2^{|P|}$. In fact we do not assume that there is a single most abstract supremum but rather the structure could have multiple maximal elements (thus making it a meet semi-lattice rather than a pure lattice).

For convenience, we will treat the abstraction function $f$ for a given lattice as invertible and use $f_P^{-1}(\mathcal{M})$ to represent the unique concrete node in the lattice that could have been abstracted (by projecting out $P$) to generate $\mathcal{M}$. We will refer to $f_P^{-1}(\mathcal{M})$ as the concretization of $\mathcal{M}$ for $P$. Figure 1 presents a simple conceptualization of an abstraction lattice for the rover domain. The edges in the lattice correspond to projecting out the presence of rocks or soil samples. Earlier works have used such abstraction lattices to estimate the user's level of understanding of the given task, by searching for the level of abstraction where an incorrect alternative raised by the user (or foil) could be supported.

## 3 Our Approach

Before we start discussing the technical details of our approach, let us look at a possible explanatory scenario.

**Example 1.** Consider the following scenario where a rover is tasked with collecting a rock sample and a soil sample from the region illustrated in Figure 2. The rover can only traverse the region via the waypoints marked on the map and its maneuverability is affected by the conditions of the terrain. The rover cannot easily traverse the region between P3 and P4 without special precautions as the region is quite rocky. Suppose a mission control operator is also keeping track of the rover's plan but may not have access to a map with the

same level of fidelity or may have incomplete knowledge of the rover's capabilities. The rover reports to the mission controller that in fact the task can not be solved. The mission control operator is confused by the rover's response and could even ask

*"Why don't you collect the rock sample from P4 and Soil sample from P7?"*

Here if the rover wants to explain the reason as to why it couldn't achieve the goal a possible way would be to clarify that certain parts of the map are hard to traverse (particularly the region around the rock sample) and because of this issue it can never reach the location of the rock sample. Thus the explanation in this case consist of two distinct parts, information about the problem (i.e traversability of certain paths) and the required subgoal that can no longer be achieved in the light of this new information. In the proceeding sections, we will layout our framework and discuss how we could leverage it to generate such explanations.

The input to our approach thus includes an unsolvable problem $\mathcal{M}_R = \langle F_R, A_R, I_R, G_R \rangle$ (in the above example this would correspond to the complete rover model) and an abstraction lattice $\mathbb{L}_{\mathcal{M}_R,P} = \langle \mathbb{M}, \mathbb{E}, \ell \rangle$, where $\mathbb{M}$ represents the space of possible models that could be used to capture the human's understanding of the task (i.e by assuming the user may or may not be aware of the fluents in the set $P \subseteq F_R$). In Example 1, $P$ could include fluents related traversability of various paths or fluents related to various rover capabilities. Given this setting, our method for identifying explanations, includes the following steps

- Identify the level of abstraction at which the explanation should be provided (Section 3.1)

- Identify a sequence of necessary subgoals for the given problem that can be reasoned about at the identified level of abstraction (Section 3.2)

- Identify the first unachievable subgoal in that sequence (Section 3.3)

Intuitively, one could understand the three steps mentioned above as follows. First, identify the level of detail at which unsolvability of the problem needs to be discussed. The higher the level of abstraction, the easier the user would find it to understand and reason about the task, but the level of abstraction should be detailed enough that the problem is actually unsolvable there. In most cases, this would mean finding the highest level of abstraction where the problem is still unsolvable.

Now even if the system was to present the problem at this desired level of abstraction, the user may be unable to grasp the reason for unsolvability. Again, our method involves helping the human in this process by pointing out a necessary subgoal (i.e., any valid solution to that problem must achieve the subgoal) that can't be achieved at the current abstraction level. Thus the second point relates to the challenge of finding a sequence of subgoals (defined by state fluents present at the explanatory level) for a given problem. In the third step, we try to identify the first subgoal in the sequence that is actually unsolvable in the given level.

Given our approach, the final explanatory message provided to the user would include model information that brings their understanding of the task to the required level and information on the specific subgoals (and previous ones that need to be achieved first) that can no longer be achieved. In cases where the unachievable subgoals are hard to understand formulas or large disjunctions, we can also use these subgoals to produce exemplar plans in the more abstract models and illustrate their failures alongside the unachievable subgoals.

## 3.1 Identifying the Minimal Level of Abstraction Required for Explanation

Following [Sreedharan, Srivastava, and Kambhampati, 2018], we will assume that the human's understanding of the task could be approximated by a model $\mathcal{M}_H = \langle F_H, A_H, I_H, G_H \rangle$, such that, the model is part of the abstraction lattice ($\mathcal{M}_R \sqsubset \mathcal{M}_H$ and $\mathcal{M}_H \in \mathbb{M}$). While the earlier work is able to use alternative plan provided by the user to identify the human model, we instead use the fact that the user expected the problem to be solvable to identify $\mathcal{M}_H$, i.e., $\exists \pi, \pi(I_{\mathcal{M}_H}) \models_{\mathcal{M}_H} G_{\mathcal{M}_H}$.

We now need to abstract this human model to a level where the problem is unsolvable (i.e the explanation level) by providing information about a certain subset of fluents previously missing from the human model (i.e information on their truth values in the initial and goal state, and how they affect various actions etc...). In the case of Example 1, this would include information on whether various paths are traversable and how the traversability of a path is a precondition for the robot to move across it. We will refer to the set of fluents that the human needs to be informed about as explanatory fluents ($\mathcal{E}$) and for Example 1, it will be $\mathcal{E} = \{can\_traverse(?x, ?y)\}$.

**Definition 1.** *Given a human model $\mathcal{M}_H$, we define a set of propositions $\mathcal{E}$ to be **explanatory fluents** if $f_{\mathcal{E}}^{-1}(\mathcal{M}_H)$ is unsolvable, i.e, $|\Pi_{f_{\mathcal{E}}^{-1}(\mathcal{M}_H)}| = 0$.*

Unfortunately, this is not an operational definition as we do not have access to $\mathcal{M}_H$. Instead, we know that $\mathcal{M}_H$ must be part of the lattice, and thus there exists a subset of the maximal elements of the lattice (denoted as $\mathbb{M}^{abs}$) that is more abstract than $\mathcal{M}_H$. In this section, we will show how the explanatory fluents for models in this subset of $\mathbb{M}^{abs}$ would satisfy $\mathcal{M}_H$ as well.

The first useful property to keep in mind is that if $\mathcal{M}_1$ is more concrete than $\mathcal{M}_2$ then the models obtained by concretizing each model with the same set of fluents would maintain this relation (although they may get concretized to the same model), i.e.,

**Proposition 1.** *Given models $\mathcal{M}_1$, $\mathcal{M}_2$ and a set of fluents $\epsilon'$, if $\mathcal{M}_1 \sqsubseteq \mathcal{M}_2$, then $f_{\epsilon'}^{-1}(\mathcal{M}_1) \sqsubseteq f_{\epsilon'}^{-1}(\mathcal{M}_2)$.*

Next, it can be shown that any given set of explanatory fluents for an abstract model will be a valid explanatory fluent set for a more concrete model

**Proposition 2.** *Given models $\mathcal{M}_1$, $\mathcal{M}_2$, if $\mathcal{M}_1 \sqsubseteq \mathcal{M}_2$, then any explanation $\mathcal{E}$ for $\mathcal{M}_2$ must also be an explanation for $\mathcal{M}_1$.*

To see why this proposition is true, let's start from the fact that $f_{\mathcal{E}}^{-1}(\mathcal{M}_1) \sqsubseteq f_{\mathcal{E}}^{-1}(\mathcal{M}_2)$ and therefore $\Pi_{f_{\mathcal{E}}^{-1}(\mathcal{M}_1)} \subseteq \Pi_{f_{\mathcal{E}}^{-1}(\mathcal{M}_2)}$. From the definition of explanation we know that

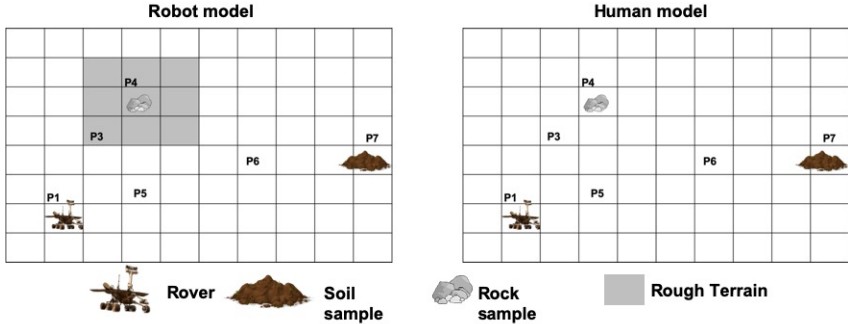

Figure 2: The map for the rover mission planning problem. The rover is required to collect a rock sample and a soil sample and then return to the original position P1. One of the rock samples is located in rough terrain (gray) that can not be traversed by the rover. The mission control operator who is monitoring the plan is currently unaware of this detail.

the concretization with respect to explanatory fluents would render the problem unsolvable (i.e $|\Pi_{f_{\mathcal{E}}^{-1}(\mathcal{M}_2)}| = 0$) and thus $|\Pi_{f_{\mathcal{E}}^{-1}(\mathcal{M}_1)}|$ must also be empty and hence $\mathcal{E}$ is an explanation for $\mathcal{M}_1$.

**Definition 2.** *Given an abstraction lattice $\mathbb{L}$, let $\mathbb{M}^{abs}$ be its maximal elements. Then the **minimum abstraction set** is defined as $\mathbb{M}_{min} = \{\mathcal{M}|\mathcal{M} \in \mathbb{M}^{abs} \wedge |\Pi_{\mathcal{M}}| > 0\}$.*

Note that for any model $\mathcal{M}_1 \in \mathbb{M}_{min}$, $\mathcal{M}_H \sqsubseteq \mathcal{M}_1$, this means by Proposition 2, any explanation that is valid for models in $\mathbb{M}_{min}$, should lead $\mathcal{M}_H$ to a node where the problem is unsolvable. Now we can generate the explanation (even the optimal one) by searching for a set of fluents that when introduced to the models $\mathcal{M} \in \mathbb{M}_{min}$ will render the problem $f_{\mathcal{E}}^{-1}(\mathcal{M})$ unsolvable. In this work, we will mostly consider the use of uniform cost search to find the least costly set of explanatory fluent, where the cost of each fluent reflects the cost of communicating information about a particular fluent. In this case, the search state consists of sets of models (with $\mathbb{M}_{min}$ being the initial state), the actions consist of the various fluent concretizations, the edges of the lattice define the successor functions and the goal test involves verifying whether the problem is solvable in each possible model in the current state.

## 3.2 Generating Subgoals of a Given Problem

Note that it would be hard to identify non-trivial subgoals for the given problem in the node at which the problem was found to be unsolvable (i.e $f_{\mathcal{E}}^{-1}(\mathbb{M}_{min})$) since there are no valid plans in that model. Fortunately, we can use models more abstract than $f_{\mathcal{E}}^{-1}(\mathbb{M}_{min})$ to generate such subgoals. We will use planning landmarks [Hoffmann, Porteous, and Sebastia, 2004] extracted from $\mathcal{M}$, where $|\Pi_{\mathcal{M}}| > 0$, as subgoals. Intuitively, state landmarks (denoted as $\Lambda = \langle \Phi, \prec \rangle$) for a model $\mathcal{M}$ can be thought of as a partially ordered set of formulas, where the formulas and the ordering need to be satisfied by every plan that is valid in $\mathcal{M}$. We will only be considering sound orderings (c.f [Richter, Helmert, and Westphal, 2008]) between landmarks, namely, (1) *natural orderings* ($\prec_{nat}$) - $\phi \prec_{nat} \phi'$, then $\phi$ must be true before $\phi'$ is made true in every plan, (2) *necessary orderings* ($\prec_{nec}$) - if $\phi \prec_{nec} \phi'$ then

$\phi$ must be true in the step before $\phi'$ is made true every time and (3) *greedy necessary orderings* ($\prec_{gnec}$) - if $\phi \prec_{gnec} \phi'$ then $\phi$ must be true in the step before $\phi'$ is made true the first time. The landmark formulas may be disjunctive, conjunctive or atomic landmarks.

Our use of landmarks as the way to identify subgoals is further justified by the fact that logically complete abstractions conserve landmarks. Formally

**Proposition 3.** *Given two models $\mathcal{M}_1$ and $\mathcal{M}_2$, such that $\mathcal{M}_1 \sqsubseteq \mathcal{M}_2$, let $\Lambda_1 = \langle \Phi_1, \prec_1 \rangle$ and $\Lambda_2 = \langle \Phi_2, \prec_2 \rangle$ be the landmarks of $\mathcal{M}_1$ and $\mathcal{M}_2$ respectively. Then for all $\phi_i^1, \phi_j^1 \in \Phi_1$, such that $\phi_i^1 \preceq_1 \phi_j^1$ (where $\prec_1$ is some sound ordering), we have $\phi_i^2$ and $\phi_j^2$ in $\Phi_2$, where $\phi_i^1 \preceq_2 \phi_j^1$, $\phi_i^1 \models \phi_i^2$ and $\phi_j^1 \models \phi_j^2$.*

This is true because $\phi_i^2 \prec_1 \phi_j^2$ holds over all the plans that are valid in $\mathcal{M}_2$, and therefore must also hold over all plans in $\mathcal{M}_1$. Though in $\mathcal{M}_1$ these landmark instances may be captured by more constrained formulas, and additionally $\mathcal{M}_1$ may also contain landmarks that were absent from $\mathcal{M}_2$. Now if we can show that in a particular model, a landmark generated from a more abstract model is unachievable (or the ordering from the previous level is unachievable) then $\phi_*^1$ becomes $\bot$ (thereby meeting the above requirement). Thereafter, for any model more concrete than $\mathcal{M}_2$, the formula corresponding to that landmark must be $\bot$. In other words, if for any model a landmark is unachievable, then that landmark can't be achieved in any models more concrete than the current one.

So given the explanatory level, we can move one level up in the lattice and make use of any of the well established landmark extraction methods developed for classical planning problem to generate a sequence of potential subgoals for $\mathcal{M}_R$.

## 3.3 Identifying Unachievable Sub-Goals

Now we need to find the first subgoal from the sequence that can no longer be achieved in the models obtained by applying the explanatory fluents ( $f_{\mathcal{E}}^{-1}(\mathbb{M}_{min})$) which will then be presented to the user. For example, in the case of Figure 1, the unachievable subgoal would correspond to satisfying $at\_rover(5, 4)$ (marked in red in $M_4$).

It is important to note that finding the first unachievable subgoal is not as simple as testing the achievability of each subgoal at the abstraction level identified by methods discussed in section 3.1. Instead, we need to make sure that each subgoal is achievable while preserving the order of all the previous subgoals. To test this we will introduce a new compilation that allows us to express the problem of testing achievement of a landmark formula as a planning problem. Consider a planning model $\mathcal{M}$ and the landmarks $\Lambda = \langle \Phi, \prec \rangle$ extracted from some model $\mathcal{M}'$, where $\mathcal{M} \sqsubset \mathcal{M}'$. We will assume that the formulas in $\Phi$ are propositional logic formulas over the state fluents and are expressed in DNF. Each $\phi \in \Phi$ can be represented as a set of sets of fluents (i.e, $\phi = \{c_1, ..., c_k\}$ and each $c_i$ set takes the form $c_i = \{p_1, ..p_m\}$), where each set of fluents represent a conjunction over those fluents. For testing the achievability of any landmark $\phi \in \Phi$, we make an augmented model $\mathcal{M}_\phi = \langle F^\phi, A^\phi, I^\phi, G^\phi \rangle$, such that the landmark is achievable *iff* $|\Pi_{\mathcal{M}_\phi}| > 0$. The model $\mathcal{M}_\phi$ can be defined as follows: $F^\phi = F \cup F^{meta}$, where $F^{meta}$ contains new meta fluents for each possible landmark $\phi' \in \Phi$ of the form

- $achieved(\phi')$ keeps track of a landmark being achieved and never gets removed

- $unset(\phi')$ Says that the landmark has not been achieved yet, usually set true in the initial state unless the landmark is true in the initial state

- $first\_time\_achieved(\phi')$ Says that the landmark has been achieved for the first time. This fluent is set true in the initial state if the landmark is already true there

The new action set $A^\phi$, will contain a copy of each action in $A$. For each new action corresponding to $a \in A$, we add the following new effects to track the achievement of each landmark

- for each $\phi' \in \Phi$ if the action has existing add effects for a subset of predicates $\hat{c}_j$ for a $c_j \in \phi'$, then we add the conditional effects $cond_1(\phi') \rightarrow \{achieved(\phi')\}$ and $cond_2(\phi') \rightarrow \{first\_time\_achieved(\phi')\}$, where $cond_1(\phi') = c_j \setminus \hat{c}_j \cup \{\hat{\phi} | \hat{\phi} \in \Phi \wedge (\hat{\phi} \prec_{nec} \phi')\} \cup \{achieved(\hat{\phi}) | \hat{\phi} \prec_{nat} \phi'\}$ and $cond_2(\phi') = cond_1(\phi') \cup \{\hat{\phi} | \hat{\phi} \prec_{gnec} \phi'\} \cup \{unset(\phi)\}$

- We add a conditional delete effect to every action of the form $first\_time\_achieved(\phi') \rightarrow (not(first\_time\_achieved(\phi')))$

The new goal would be defined as $G^\phi = \{first\_time\_achieved(\phi)\}$.

This formulation allows us to test each landmark in the given sequence and find the first one that can no longer be achieved. To ensure completeness, we will return the final goal if all the previously extracted landmarks are still achievable in $f_\mathcal{E}^{-1}(\mathbb{M}_{min})$. Now given an ordered set of landmarks, we can identify the first unsolvable landmark by testing the solvability of the $F^\phi$ for each landmark in the order they appear in the sequence.

Since the above formulation is designed for DNF, we can generate compilation for cases where the landmarks use either un-normalized formulas or CNF by converting them first into DNF formulas.

## 4 Planning Problems with Plan Advice

Let us now discuss how we could extend the methods presented in earlier sections to cases where the user provides plan advice. In such cases, the user imposes certain restrictions on the kind of solution they expect, either as an alternative to the solution the system may come up with on its own or as a guide to help the system come up with solutions when it claims unsolvability.

As pointed out in [Myers, 1996], such advice can be compiled into plan constraints in the original problem. A number of approaches have been proposed to capture and represent plan constraints [Bacchus and Kabanza, 2000; Nau et al., 2001; Kambhampati, Knoblock, and Yang, 1995; Baier and McIlraith, 2006], and each of these representational choices has its unique strengths and weaknesses. In general, we can see that these plan constraints as specify a partitioning of the space of all valid plans to either acceptable (i.e it satisfies the constraints) or unacceptable. So we can define, constraints as follows

**Definition 3.** *The partition specified by a **constraint** $\sigma$ on a given set of plans that is specified by a membership function $\sigma : \Pi \rightarrow [0, 1]$, where $\Pi$ is the set of all plans.*

We will slightly abuse notation and for a given set of plans $\hat{\Pi}$ we will use $\sigma(\Pi')$ to denote $\{\pi | \pi \in \hat{\Pi} \wedge \sigma(\pi) = 1\}$ (i.e the subset of $\Pi'$ that satisfies the constraint). If we can assume some upper bound on the possible length of plans in $\sigma(\Pi_\mathcal{M})$ (which is guaranteed when we restrict our attention to non-redundant plans for standard classical planning problems), then we can assert that there always exists a finite state machine that captures the space of acceptable plans

**Proposition 4.** *Given a constraint $\sigma$ and a model $\mathcal{M}$, there exists a finite state automaton $\mathcal{F}^{\sigma,\mathcal{M}} = \langle \Sigma, \mathbb{S}_{\mathcal{F}^{\sigma,\mathcal{M}}}, S_0, \delta, E \rangle$, where $\Sigma$ is the input alphabet, $\mathbb{S}_{\mathcal{F}^{\sigma,\mathcal{M}}}$ defines the FSA states, $S_0$ is the initial state, $\delta$ is the transition function and $E$ is the set of accepting states, such that $\sigma(\Pi_\mathcal{M}) = \mathcal{L}(\mathcal{F}^{\sigma,\mathcal{M}}) \cap \Pi_\mathcal{M}$, where $\mathcal{L}(\mathcal{F}^{\sigma,\mathcal{M}})$ is the set of strings accepted by $\mathcal{F}^{\sigma,\mathcal{M}}$.*

The existence of $\mathcal{F}^{\sigma,\mathcal{M}}$ can be trivially shown by considering an FSA that has a path for each unique plan in $\mathcal{F}^{\sigma,\mathcal{M}}$. We believe that this formulation is general enough to capture almost all of the plan constraint specifications discussed in the planning literature, including LTL based specifications, since for classical planning problems these formulas are better understood in terms of $LTL_f$ [De Giacomo and Vardi, 2015] which can be compiled into a finite state automaton.

We can use $\mathcal{F}^{\sigma,\mathcal{M}}$ to build a new model $\sigma(\mathcal{M})$ such that a plan is valid in $\sigma(\mathcal{M})$ if and only if the plan is valid for $\mathcal{M}$ and satisfies the given specification $\sigma$, i.e., $\forall \pi, \pi \in \Pi_{\sigma(\mathcal{M})}$ iff $\pi \in \sigma(\Pi_\mathcal{M})$

For $\mathcal{M} = \langle F, A, I, G \rangle$, we can define the new model $\sigma(\mathcal{M}) = \langle F_\sigma, A_\sigma, I_\sigma, G_\sigma \rangle$ as follows

- $F_\sigma = F \cup \{\text{in-state-}\{S\} | S \in \mathbb{S}_{\mathcal{F}^{\sigma,\mathcal{M}}}\}$

- $A_\sigma = A \cup A_\delta$

- $I_\sigma = I \cup \{\text{in-state-}\{S_0\}\}$

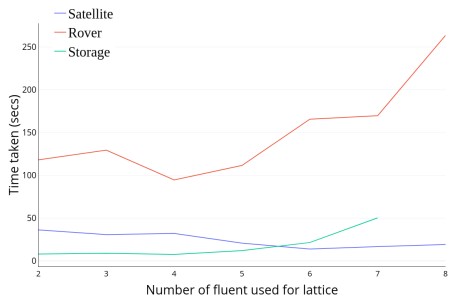

Figure 3: The graph compares the time taken to generate the explanation for three of the domains for increasing size of lattices.

- $G_\sigma = G \cup \{\text{in-state-}\{S\}|S \in E\}$

$A_\delta$ are the new meta actions responsible for simulating the transitions defined by $\delta : \mathbb{S}_{\mathcal{F}^\sigma, \mathcal{M}} \times \Sigma \rightarrow pow(\mathbb{S}_{\mathcal{F}^\sigma, \mathcal{M}})$. For example, if $\delta(S_1, a) = \{S_1, S_2\}$, where $a$ corresponds to an action in $A$, then we will have two new actions $a^1_{S_1, a} = \langle prec^a \cup \{\text{in-state-}\{S_1\}\}, adds^a \cup \{\text{in-state-}\{S_2\}\}, dels^a \cup \{\text{in-state-}\{S_1\}\}\rangle$ and $a^2_{S_1, a} = \langle prec^a \cup \{\text{in-state-}\{S_1\}\}, adds^a, dels^a\rangle$. In cases like LTL, the FSA state transitions may be induced by the satisfaction of some formula, so the new meta action may have preconditions corresponding to that formula, with no other effects but changing the fluent corresponding to the state transition.

The above formulation merely points out that there always exists a way of generating $\sigma(\mathcal{M})$ from the given specification $\sigma$ and $\mathcal{M}$. For many constraint types, there may exist more efficient ways of generating models that satisfy the requirements of $\sigma(\mathcal{M})$.

Once we have access to $\sigma(\mathcal{M})$, we should be able to use the methods discussed in earlier sections to explain unsolvability of $\sigma(\mathcal{M})$ and hence why the constraint isn't feasible. To facilitate such explanation, we will build an abstraction lattice for the constrained problems $\mathbb{L}_{\sigma\mathcal{M}, P}$ such that $P \cap (F^\sigma \setminus F) = \phi$, i.e the abstraction lattice only affects the fluents from the original problem and not the new ones introduced as part of the compilation. In fact, we can induce such a lattice by considering the lattice generated for the original problem and then replacing each node in the lattice with the corresponding compiled problem, to see why this would induce a valid abstraction lattice, consider the following property

**Proposition 5.** *Given models* $\mathcal{M}_1$, $\mathcal{M}_2$ *and a constraint specification* $\sigma$, *if* $\mathcal{M}_1 \sqsubseteq \mathcal{M}_2$, *then* $\sigma(\mathcal{M}_1) \sqsubseteq \sigma(\mathcal{M}_2)$.

To see why this is true, just assume that the reverse was true, that $\sigma(\mathcal{M}_2)$ is not a logically complete abstraction of $\sigma(\mathcal{M}_1)$. This means that there are plans in $\Pi_{\sigma(\mathcal{M}_1)}$ that are not part of $\Pi_{\sigma(\mathcal{M}_2)}$. From the definition of $\sigma(\mathcal{M}_2)$, we know that $\Pi_{\sigma(\mathcal{M}_2)} = \Pi_{\mathcal{M}_2} \cap \mathcal{L}(\mathcal{F}^\sigma)$. If there exist a $\pi \in \Pi_{\sigma(\mathcal{M}_1)}$, such that $\pi \notin \Pi_{\sigma(\mathcal{M}_2)}$, then $\pi \notin \Pi_{\mathcal{M}_2}$. Which means $\mathcal{M}_1 \not\sqsubseteq \mathcal{M}_2$, hence contradicting our assumptions.

Revisiting Example 1, the question asked by the user could be seen as an advice, where the corresponding constraint covers all plans where the rover performs the actions collect_rock_sample P4, collect_soil_sample P7, irrespective of the exact position and order in which the actions appears in the plan. More generally, we could think of this plan advice as being a special case of advice where the user wants to ensure presence of certain actions in the plan with some partial ordering among them (eg: "Why don't you pickup block B and then C?", "Make sure that you have cleared Room1 before you move on to Room2 and Room3" etc..). Such advice could be represented as partial plans [Kambhampati, Knoblock, and Yang, 1995], which in general can be captured as a partially ordered multiset of the form[1] $\hat{\pi} = \langle \hat{A}, \leqslant \rangle$, where $\hat{A}$ is a multiset over grounded actions and $\leqslant$ defines ordering constraints over these grounded actions. A plan $\pi = \langle a_1, ..., a_n \rangle$ is said to be a candidate plan for the given partial plan $\hat{\pi}$, if there exists a mapping function $\mu : \hat{A} \rightarrow [1, |\pi|]$ that maps each action in $a \in \hat{A}$ to a position in the plan such that $a = a_{\mu(a)}$ and if $a < b$ for $a, b \in \hat{A}$, then $\mu(a) < \mu(b)$. Such partial plans can be fairly easily compiled into a classical planning model (such that it satisfies $\sigma(\mathcal{M})$) by extending the compilation methods discussed in [Ramírez and Geffner, 2010], without relying on an intermediate finite state machine.

The corresponding partial plan for the question specified above would be

$\hat{\pi} = \langle \{\text{collect\_rock\_sample P4, collect\_soil\_sample P7}\}, \rangle$

Let us assume that in this case the observer could be unaware of certain domain constraints such as the rover's inability to traverse certain regions on the map the fact that not all rovers are capable of collecting rock and soil samples or that they are not always equipped to store these samples. In this case, possible user models can be captured by a lattice build using the following fluents $P = \{$(can_traverse ?x ?y), (equipped_for_soil_sample ?r), (equipped_for_rock_sample ?r), (store_of ?r)$\}$. Now our approach would identify the user need to be made aware of the fact that not all regions of the map are equally traversable (i.e inform the user about can_traverse ?x ?y) and how its a precondition for move action), furthermore given this property the robot can no longer reach the waypoint 4 which is required to complete this task (i.e the unreachable landmark is (at rover0 waypoint4)).

## 5 Evaluations

### 5.1 User Studies

Our first concern with evaluating explanations based on landmarks was to establish that they constitute meaningful explanations for naive users. As a simple alternative to our explanations, we consider providing to the user a set of potential solutions (generated from a higher level of abstraction) and their individual failures. For the study, we recruited around 120 master turkers from Amazon's Mechanical Turk and tested the following hypotheses

- **H1** - Users prefer concise explanations over ones that enumerate a set of possible candidates for a given piece of plan advice

[1]We are presenting a simplified definition of a partial plan. The full definition allows for the representation of more complex constraints than mere ordering constraints, such as contiguity constraints and interval protection constraints.

| Domain Name | $|P|$ | Average Runtime (secs) | Explanation Cost | Cost of explaining $M_R$ |
|---|---|---|---|---|
| Blocksworld | 4 | 8.141 | 11.6 | 30.2 |
| Satellite | 8 | 19.15 | 6 | 43.6 |
| Depots | 5 | 20.229 | 13 | 51 |
| Rover | 8 | 263.635 | 7.5 | 15.75 |
| Storage | 7 | 50.348 | 20 | 55.8 |
| Over-Rover* | 8 | 2047.360 | 29.8 | 92.6 |
| Over-tpp* | 8 | 1065.542 | 842.8 | 881.2 |
| Bottleneck* | 3 | 504.431 | 60.8 | 66.2 |

Table 1: Table showing runtime for explanations generated for standard IPC domains. The explanation costs capture the number of unique model updates (changes in effects/precondition etc..) corresponding to each explanation

- **H2** - Users prefer concise explanations that contain information about unachievable landmarks over ones that only show the failure of a single exemplary plan

For the hypotheses, we presented the study participants with a sample dialogue between two people over a logistics plan to move a package from one location to another. The dialogue included a person (named Bob) presenting a plan to another (named Alice), and Alice asks for an alternative possibility (i.e specifies a constraint on the solution). Now the challenge for Bob is to explain why the constrained problem is unsolvable. For example, in one example Bob presents a rather convoluted plan that involves the package being transferred through multiple trucks to a train and then to the final destination. This leads to Alice asking the package to be delivered via an airplane.

For H1, in addition to some model information that Bob was unaware of, the potential explanations included either (a) the information on the unachievable landmark, (b) landmark information with the failure details of a specific exemplary plan or (c) a set of plans that satisfy the constraints and their corresponding failures. For the earlier example this meant Bob explains to Alice the limited availability of Truck fuel and (a) the impossibility of getting the package to the airport or (b) the the impossibility of getting the package to the airport and a specific plan (eg: *truck1 picks up package moves to location two then to three ...*) along with its point of failure (eg: *truck1 runs out of fuel when it reaches location three*) or (c) three example plans involving various trucks trying to get the package to the airport and their specific points of failures (each of which fails at different steps but before reaching the airport).

For this study, we used 45 participants and each participant was assigned one of three possible maps for each hypothesis and was paid $1.25 for 10 mins. We used a control question to filter participant responses, so as to ensure their quality. Out of the 39 remaining responses, we found 94.8% of users chose to select the more concise explanation (i.e (a) or (b)), and 51.28% of the users chose explanations that involved just landmarks.

For H2, we used 75 participants and presented each participant with explanations that include (a) just landmark information, (b) landmark information with failure details of an exemplary plan and (c) just the exemplary plan failure. Here participants were paid $1 for 10 mins for H2 as the explanatory options were much simpler. After filtering using the control question, we found that out of 60 valid entries 75.4% of participants preferred explanations that included landmark information ((a) or (b)) and 44.2% wanted both landmarks and exemplary plan (i.e (b)). The supplementary file at http://bit.ly/2HQ5sTv contains more details on the study setup.

## 5.2 Empirical Studies

In this section, we will present the results of an empirical evaluation of the computational characteristics of our approach. One big concern with the methods discussed in this work is the fact that they involve solving multiple planning problems. Thus we were interested in identifying the runtime for generating explanations on a set of standard planning benchmarks.

To evaluate our methods, we considered eight planning domains and chose five problem instances for each of the domains. For each domain, we used a subset of the domain predicates to generate the abstraction lattice (i.e we set the subset as the set of fluents $P$ used to define the lattice). The first five domains and their problem instances consisted of standard IPC domains and problem instances used in previous IPC competitions [International Planning Competition, 2011]. Each problem instance was made unsolvable by including plan constraints that avoid a specific landmark of the original problem. The constraints were coded using domain control programs [Baier, Fritz, and McIlraith, 2007] of the form
```
while ¬φ ∧ ¬(goal_completed)
do   any
done
```
Where $\phi$ is the landmark formula and (goal_completed) is the goal fluent (generated by a new goal_action whose preconditions are the original goals of the problem). The constraints ensure that any valid plan must avoid the landmark $\phi$ and thereby rendering it unsolvable. The next three domains were selected from the set used for the 2016 unsolvability competition [Unsolvability International Planning Competition, 2016] (these domains are marked with an asterix in the results table). All instances were run with a timeout of 100 minutes (all problems were solvable under this time limit) and all landmarks were generated using the fast-downward implementation of [Keyder, Richter, and Helmert, 2010] (where we set the subset sizes to one for the first five domains and to two for the rest).

Table 1 presents the results of our tests on these domains. It shows the number of fluents used to generate the lattice ($|P|$), the average runtime, the cost of the generated explanations and the cost of presenting the most concrete model to the user. For each scenario, we created a complete lattice for all the fluents considered for abstraction (i.e $|\mathbb{M}| = 2^{|P|}$). The cost of the explanation captures the amount of information to be provided to the user as part of the explanation. This could include information regarding the various explanatory fluents and is here captured roughly by the number of places within the domain definition where these fluents appear. The cost also reflects the inferential overhead demanded from the user (since providing more information translates to the user needing to understand the domain at a much more concrete level).

For a sample explanation, consider the overconstrained rover domain, where the rovers' actions are limited by their energy levels and the energy of the rover isn't enough to finish the task. In one of the instances where the rover energy level is at 33 and the original problem had a goal consisting of eight propositions (each referring to the need for communicating a particular soil sample, rock sample or sending an image for different objectives), our approach was able to identify that the user needs to understand fluents related to energy ((energy ?x ?y) and (energycost ?x ?y ?z)) and identified two subgoals out of the eight that it could not achieve.

Figure 3 presents the variations in average runtime for three of the domains as the size of the lattice were increased (the X-axis represents the number of fluents that were used to build the lattice and Y the runtime in seconds). Note that, in general, the runtime increases as the lattice size increase due to the increase in the search space, but in all three domains there are points where the runtime decrease when the lattice size increases. This is expected since with an increase in the size of lattice, the planning problems whose unsolvability are being tested becomes simpler.

## 6 Related Work

As discussed earlier, our methods for identifying the level of explanations are based on the expertise level modeling approaches introduced in [Sreedharan, Srivastava, and Kambhampati, 2018]. These two works are quite closely connected and in fact, the contrastive explanations of the type studied in the earlier paper, where the user presents alternative plans (i.e the foils for the explanations) that are then refuted by the system, is a special case of our approach for handling problems with plan advice. The problems studied in that earlier paper can be thought of as capturing cases where the advice only allows for a single plan. Also, one could argue that people would be more comfortable giving advices as foils rather than full plans. Part of our explanations also try to reveal to the user information about the current task that was previously unknown to them. Thus our methods could also be understood as an example of explanation as model-reconciliation [Chakraborti et al., 2017]. Since our methods use abstractions, our approach doesn't make too many demands on the inferential capabilities of the user and hence can be applied to much larger and more complex domains.

Another closely related direction has been the work done on explaining unsynthesizability of hybrid controllers for a given set of high-level task specifications [Raman and Kress-Gazit, 2013]. The work tries to identify the subformulas of the given specification that lead to the unsynthesizability. This particular approach is specific to the planning framework detailed in [Finucane, Jing, and Kress-Gazit, 2010] and the objective of the work parallels the goals of work like [Göbelbecker et al., 2010].

Outside of explanation generation, the work done in the model checking community is closely related to our current problem [Grumberg and Veith, 2008]. In fact, the hierarchical approach to identifying a model that can invalidate the given foil specification, can be seen as a special case of the CEGAR based methods studied in the model-checking community [Clarke et al., 2000]. Most work in this field focuses on developing methods for identifying whether a given program meets some specifications and failures to meet specification are generally communicated via counterexamples.

Another related problem is that of identifying whether a given problem is unsolvable. In our setting, we assume that the system is capable of correctly identifying whether a given problem is unsolvable or not and in general this can be a time consuming process. Thankfully the problem of efficiently identifying whether a given planning problem is unsolvable is an active research area (cf. [Steinmetz and Hoffmann, 2017; Kolobov, Weld, and others, 2010]) and solutions to this problem can be easily leveraged by our approach to improve the overall efficiency of the system.

## 7 Conclusion and Future Directions

The work presented in this paper investigates the problem of generating explanations for unsolvability of a given planning problem. We also saw how the same methods apply when dealing with problems with plan constraints. In addition to extending these methods to more expressive domains, an interesting extension would be to try tackling cases where the current problem is solvable but all the solutions are too expensive. While this additional cost threshold could be seen as a constraint, the setting becomes a lot more interesting when the action costs are affected by the abstractions (c.f state dependent costs [Geißer, Keller, and Mattmüller, 2016]). With respect to contrastive explanations, this would correspond to cases where the alternative posed by the user is more expensive than the plan proposed by the robot. Finally, an obvious challenge to fully realize this method in practical scenarios is to develop methods to convert user questions to plan constraints. Methods like [Tenorth et al., 2010] can be used to convert natural language statements to constraints like partial plans. Expert users can also directly write LTL and procedural programs as a way of interrogating the system.

## Acknowledgments

This research is supported in part by the ONR grants N00014-16-1-2892, N00014-18-1-2442, N00014-18-1-2840, the AFOSR grant FA9550-18-1-0067, the NASA grant NNX17AD06G, NSF grant 1844325 and a JP Morgan AI Faculty Research grant.

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
