# OpenReview forum: "Why Can't You do that HAL? Explaining Unsolvability of Planning Tasks"
_icaps-conference.org/ICAPS/2019/Workshop/XAIP — XAIP 2019_

### Official Review · AnonReviewer1 · 2019-04-24
**Interesting new approach to explaining unsolvability**

**Rating:** 5
**Confidence:** 3

**Review:**

The paper introduces a novel way of explaining why a planning problem is unsolvable, via a combination of choosing an abstraction level (in a hierarchy of projections: the maximally abstract version which is still unsolvable) with generating the first unsolvable landmark at that level (landmarks being taen from the maximally concrete solvable version). The paper conducts first user studies and runs some computational experiments.

Overall, this is a nice contribution at a mature stage. Definitely an accept for XAIP. The issues I see are:

The introduction makes much too broad claims about the originality of the problem addressed. The authors pretend that this is the first work aiming at explaining unsolvability to humans. However, the 2010 work by Goebelbecker et al had exactly the same aim. It's got "excuse" in the title but says "explanation" already in the abstract. In any case, this is the exact same objective. The techniques suggested are quite different, and so everything is fine content-wise. But the presentation needs to be corrected.

Furthermore, although the abstraction part of this work relates very closely to previous work by Chakraborti et al on model reconciliation, that work is mentioned for the first time on the last page of the paper. It must be mentioned in the introduction already.

A technical issue is that the constraint-compilation sigma(M) may not actually be defined for a more abstract model, namely if the predicates/planning-state properties referred to in the automaton's transition function are abstracted away. So Proposition 5 is flawed as stated. Actually the automaton has to be defined relative to a task to even be able to formulate this problem. In this sense, Proposition 4 also is flawed as stated. I believe this issues are fixable so don't view them as reason to reject. But they mst be fixed, and it needs to be made clear how this will be done.

It also must be made much clearer in Section 4 that, and exactly how, "the methods discussed in earlier sections" can be used.

I disagree that there is "no direct way of extracting meaningful subgoals" (introduction; statement repeated in similar way later on) just because a planning problem is unsolvable. This actually seems to me one more instance of a tendency to formulate claims too broadly. Simple counter-example: What about shared preconditions? If g is a goal and p is a precondition of all actions achieving g, then certainly p is a "meaningful subgoal", even if g is unsolvable. Right? Such necessary subgoal analysis is actually all over the place in the landmarks literature. Don't get me wrong, I like your solution to look at the first solvable abstraction layer and use standard landmark definitions there (which indeed per se rely on solvability). But you need to put your approach into the right frame. It is one possible approach. Not the only possible one.

Minor comments:

 "we can also use these subgoals to produce exemplar plans and illustrate their failures alongside the unachievable subgoals" ==> How so? These subgoals are unsolvable after all. Are you referring here to the generation of abstract plans and pointing out failures in more concrete models?

Def 3 last Pi has an M subscript missing?

"function free fragment of classical planning" does not translate to an unambiguous definition in my mind. Do you mean continuous state variables? Or fo you intent to refer to functional strips here as well? If you include statements like this, please be precise.

I'm a little uncertain about the significance of the user studies' results. Playing the devil's advocate, I would surmise that the alternatives provided to the landmarks, based on example plans, are just bad in an obvious/trivial way. The vast majorities of user votes to this effect certainly agree with that perception. Please add a few more words discussing your views/your conclusions from these studies.

---

### Official Review · AnonReviewer2 · 2019-05-08
**Recommended for acceptance**

**Rating:** 4
**Confidence:** 3

**Review:**

The paper treats an important problem of explaining why the goals are unachievable for a planning problem. Authors propose an approach of decomposition of the planning problems in subproblems by identifying unachievable subgoals and pointing them out to a user. The main idea follows from the previous work in planning on defining abstraction lattice which enables identifying abstraction models.

The contribution of the paper is indeed in the scope of XAIP and therefore recommended for acceptance.

Authors carried out the user as well as empirical studies which showed the preference of explanations for users as well as the impact of lattice size to the runtime for an explanation.
The paper is generally well organized and written. Section 5 needs some revisions and second reading. Also, the author could give more details on experiments. There are some minor comments to be corrected which can be found at the end of this review.

Some additional questions and comments:
- Figure 4 mentioned before Figures 2 and 3. Not clear in the text in section 5.1. the same sentence repeats at the end of 5.2
- Figure 2- it is not clear how Explanation Cost is calculated. It should be defined in the text as well.
- maybe the algorithm would be better instead of a partial program in  section 5.2
- rover couldn't follow-> rover could not follow
- Section 6.1 H1- paragraph - some repetitions
- The link for the supplementary file does not work

---

### Decision · Program_Chairs · 2019-05-15

**Decision:**

Accept

**Comment:**

The reviewers agree to accept. Please address all review criticism as best possible for the final paper version and its presentation at the workshop. Looking forward to discuss your work at the workshop!